# PROSE: POINT RENDERING OF SPARSE-CONTROLLED EDITS TO STATIC SCENES

## ABSTRACT

Advances in neural rendering have enabled high-fidelity multi-view reconstruction and rendering of 3D scenes. However, current approaches of free-form shape editing can result in inaccurate and imprecise edits due to the need for proxy geometry or yield surface discontinuities in large deformations. In this work, we present a novel method based on a point-based neural renderer, PAPR (Zhang et al., 2023), that addresses both issues – no proxy geometry needs to be fitted and surface continuity is preserved after editing. Specifically, we design a novel way to guide shape editing with a set of sparse control points. We demonstrate that our method can effectively edit object shapes while preserving surface continuity and avoiding artifacts. Through extensive experiments on both synthetic and real-world datasets with various types of non-rigid shape edits, we show that our method consistently outperforms existing approaches.

## 1 INTRODUCTION

Photo-realistic rendering and free-form shape deformation of 3D models are essential in animation, design, and gaming. Recent advances in neural rendering (Mildenhall et al., 2020; Müller et al., 2022; Xu et al., 2022; Kerbl et al., 2023; Zhang et al., 2023) have enabled high-fidelity reconstructions of 3D scenes. However, to obtain these reconstructions, the 3D scenes must exist in the real world. How do we create derivatives of these scenes that exist only in our imagination? One approach is to perform *zero-shot* deformation of reconstructed scenes. There are two main challenges: (1) enabling low-effort yet precise controls for deformation, and (2) preserving rendering quality after deformation. The two challenges are often intertwined. For example, naïvely changing the 3D representation at just a few points can result in discontinuous surfaces and unrealistic rendered images, whereas requiring the user to define a complete deformation field is too laborious to be practical.

To tackle these challenges, existing approaches parameterize the neural scene representation with proxy geometry such as a cage (Xu & Harada, 2022; Peng et al., 2022; Jambon et al., 2023; Huang & Yu, 2024) or a mesh (Yuan et al., 2022; Liu et al., 2023; Wang et al., 2023; Zhou et al., 2023; Yang et al., 2022; Gu'edon & Lepetit, 2024; Jiang et al., 2024; Waczy'nska et al., 2024; Gao et al., 2024a;b), and then deform the scene by editing the proxies. However, such process often introduces errors and additional points of failure when converting to and from the proxy geometry and is limited to edits the proxy geometry can support. Constructing a high-quality cage could be a complex and time-consuming task, as the cage needs to tightly enclose the geometry with a reasonably low number of vertices. Furthermore, the cage must be free of self-intersections, yet it can still become self-intersecting after editing, resulting in erroneous deformations. For the mesh-based deformation methods, they either map the sampled points in the deformed space back to a canonical space after deformation, or directly binding elements like Gaussians to the mesh. In either cases, the rendering quality heavily relies on the mesh's accuracy, which often necessitates additional modeling work to produce. Moreover, part-level manipulations, such as rotating a car wheel, are challenging when editing a cage or mesh due to the connectivity constraints.

In this work, we develop a novel approach that avoids these limitations. In particular, our approach does not require fitting a geometry proxy. Instead, it allows the user to directly manipulate the neural scene representation and can handle out-of-distribution edits. To achieve this, we use point-based representations (Xu et al., 2022; Kerbl et al., 2023; Zhang et al., 2023), which have gained significant attention for their high fidelity and efficiency in representing geometry. These methods represent

Figure 1: Overview of the PROSE pipeline. PROSE allows users to deform the scene by manipulating a few of sparse control points, depicted in orange in the figure. Our approach allows for flexible scene editing such as twisting, stretching, and squeezing, while preserving surface continuity, avoiding distortions, and preventing the introduction of gaps or unwanted artifacts in the edited regions.

a 3D shape with a set of discrete points. Hence to edit the 3D shape, the user only needs to edit the affected points. While this significantly simplifies the editing process, care needs to be taken to avoid introducing surface discontinuities caused by gaps between points. For example, to deform 3D Gaussian splatting-based (3DGS) representations, one must adjust the covariances of every affected splat appropriately, which is challenging. To address this, existing methods often bind the splats to proxies following heuristic rules. However, the use of proxies introduce issues. For example, GaMeS (Waczy'nska et al., 2024) and Mani-GS (Gao et al., 2024b) do not constrain Gaussians to remain inside the mesh triangles, so the Gaussians may protrude beyond surface during deformation. Mesh-GS (Gao et al., 2024a) allows each Gaussian to shift along the normal direction of its associated triangle face, which can lead to rendering artifacts and non-faithful edits. Unlike these methods, SC-GS (Huang et al., 2023) avoids geometry proxies and instead supports editing with a sparse set of control points. However, it does not adjust the sizes of deformed Gaussians, which leads to visible gaps and holes in cases of large deformations as mentioned above.

A key challenge in editing 3DGS is that the scene is represented by discrete, anisotropic Gaussians, each covering a patch of the surface. Deforming these patches requires a coherent transformation of each Gaussian, for example, bending a Gaussian (which is impossible as it's no longer a Gaussian). Without doing so, visual discontinuities like gaps and tears can happen. To get around this conundrum, we choose a different point-based representation, PAPR (Zhang et al., 2023). Unlike Gaussian splatting, PAPR does not use splats; it only models the center of each point without spatial extent. To fill the gaps between points, PAPR trains a renderer which interpolates between points using attention weights learned from local point configurations around each ray. For the purposes of shape deformation, PAPR offers two benefits: (1) the user only needs to move points and does not worry about adjusting covariances, and (2) unlike with Gaussians, non-rigid deformation of points always results in a valid set of points. Therefore, PAPR's scene representation is both editable with little effort and able to generalizes well to shape deformations.

However, even though PAPR has fewer parameters to modify than the Gaussian splats, it still consists of tens of thousands of points. Editing all of them manually can be laborious. To address this, we use a sparse set of control points to move all free points without constructing any sophisticated geometry proxy. However, naïve approaches can result in free points moving erratically when control points are adjusted. Since the neural renderer has not seen such unusual arrangements of free points during training, rendering quality can degrade as a result. In this work, we address this issue by designing a method that jointly optimizes rendering quality and the way free points move as a function of sparse control point positions. Our method can be adapted to various deformation strategies as long as they are differentiable. As a proof-of-concept, we demonstrate its effectiveness with a simple deformation strategy, and the performance of the method shows its overall promise more broadly.

**Contributions**   Below are our key contributions:

- We propose leveraging PAPR to eliminate the need for the construction of geometry proxies in joint neural rendering and shape deformation pipelines.
- Moreover, we develop a method that enables editing PAPR's representation with sparse controls while maintaining high-fidelity appearances.
- We compare to the state-of-the-art neural renderers supporting shape deformation, and show a significant improvement in results.

## 2 RELATED WORK

### 2.1 3D SHAPE DEFORMATION

There is a long line of work in geometry processing on editing 3D shapes (Gain & Bechmann, 2008; Yuan et al., 2021), which generally assumes that 3D shapes are provided as input. Among these works, surface-based methods use parametric patches or surface meshes as proxies to manipulate shapes (Feng et al., 1996; 2006; Jin & Li, 2000; Decaudin, 1996; Kobayashi & Ootsubo, 2003; Singh & Fiume, 1998; Ju et al., 2005; Angelidis et al., 2004; Von Funck et al., 2006), popular approaches include Laplacian (Gao et al., 2019; Lipman et al., 2005; Sorkine & Alexa, 2007; Sorkine et al., 2004; Sorkine, 2005) and cage-based (Ju et al., 2005; Yifan et al., 2020; Zhang et al., 2020) methods. However, designing an appropriate proxy (e.g., mesh or cage) that closely fits the target model is challenging. Manual creation can be time-consuming and may require extensive expertise, while automatic generation methods might not always produce optimal proxies, especially for complex or highly detailed models. Point-based deformation methods (Borrel & Bechmann, 1991; Borrel & Rappoport, 1994; Bechmann & Dubreuil, 1993; 1995; Aubert & Bechmann, 1997; Raffin et al., 2000; Bechmann & Gerber, 2003; Hsu et al., 1992; Hu et al., 2001; Lee et al., 1995; Moccozet & Magnenat-Thalmann, 1997; Ruprecht & Müller, 1993; Botsch & Kobbelt, 2005), on the other hand, manipulate shape with a set of freely positioned points, offering greater flexibility. Overall, these methods assume that the 3D objects are given, and primarily aim to preserve certain geometric properties of the 3D objects during deformation. Orthogonal to these methods, our goal is to optimize the rendering quality of the deformation with a given deformation technique.

More recent approaches (Yuan et al., 2022; Xu & Harada, 2022; Peng et al., 2022; Jambon et al., 2023; Wang et al., 2023; Zhou et al., 2023; Yang et al., 2022) edit the scene representations learned from 2D images. To edit scenes represented by implicit representations or neural fields (Mildenhall et al., 2020), these methods often convert them into meshes (or cages) as proxy geometry, which is to be edited and then converted back to the original scene representations. The conversion to proxy geometry and back, however, introduces approximation errors and additional points of failure, because some geometric features that can be easily represented in the original representation cannot be in proxy geometry, and vice versa. Moreover, the edits that can be done are limited to the edits the proxy geometry can support, for example, part-level editing could be challenging with a mesh or cage due to the connectivity constraints, such as to spin the wheels of a car. Unlike these methods, our method does not need to construct or convert to proxy geometry.

Another line of work (Zheng et al., 2022; Huang et al., 2023; Wu et al., 2023; Bian et al., 2025; Feng et al., 2025) focuses on dynamic scenes, where observations of different shapes over time are used to train a neural scene representation that can adapt to the shape variations. They also learn to associate each shape with a set of keypoints, so that the shape can change when the keypoints are edited. However, these methods can fail to generalize to out-of-distribution (OOD) shapes that are not observed in the dynamic scenes. In contrast, our method does not require dynamic scene observations and can generalize to OOD edits, even when trained only on static scenes.

### 2.2 POINT-BASED NEURAL RENDERING

Point-based representations (Xu et al., 2022; Kerbl et al., 2023; Zhang et al., 2023) have gained significant attention in neural rendering. Some methods (Aliev et al., 2019; Rakhimov et al., 2022; Ost et al., 2021; Kopanas et al., 2021) assume a given point cloud, which can be obtained from Structure-from-Motion (SfM), Multi-View Stereo (MVS) or LiDAR. Other approaches learn the point cloud, possibly starting from an initialization. Most of them use 2D splat-based rasterization, such as (Wiles et al., 2019; Lassner & Zollhöfer, 2021; Rückert et al., 2021; Zhang et al., 2022; Zuo & Deng, 2022). Notably, unlike these methods, Kerbl et al. (2023) uses 3D Gaussian splats. While easier to edit than neural fields, these point-based methods may still have parameters that are difficult to get right, like covariances of 3D Gaussian splats. Other methods (Zhang et al., 2023; Chang et al., 2023) learn representations where each point has no spatial extent and instead use an attention mechanism (Vaswani et al., 2017) to interpolate between nearby points and fill the gaps between them. As mentioned in the introduction, we build upon PAPR to avoid having to adjust covariances or add points after editing. None of the methods in this section offer ways to edit shapes with a sparse set of control points, unlike our method.

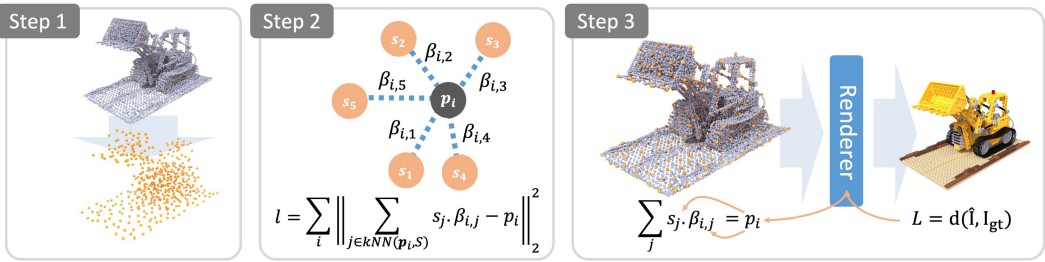

(a) 3D Gaussian Splatting        (b) Proximity Attention Point Rendering

Figure 2: As shown on the left, in 3DGS editing the points requires adjusting each Gaussian splat's covariance to eliminate gaps. However, this adjustment may introduce distortions, as circled. Instead, PAPR models only the center of each point and learns a renderer that interpolates between points using an attention mechanism. This interpolation is based on the local point configurations, for example, the relative distances between points, with many variations observed from different rays during training. This variability enables PAPR to effectively generalize to unseen point configurations and find a promising interpolation of the points after editing, as shown on the right.

Figure 3: An overview of PROSE's pipeline. **Step 1**: We extract the control points from a pretrained PAPR point cloud. **Step 2**: We represent each point in the point cloud as a linear combination of the $k$-nearest control points, where the coefficients, $\beta_{i,j} = \sigma(b_{i,j})$, are the results of minimizing the loss $l$. **Step 3**: We use the parameterized point cloud from Step 2, to jointly optimize the control point locations and the coefficients. The orange arrows indicate the gradient backpropagation.

## 3 PRELIMINARIES

PAPR learns a point-based representation of a 3D scene from multi-view RGB images and corresponding camera parameters. The point-based representation $\mathcal{P} = \{(\mathbf{p}_i, \mu_i)\}$, where $i \in \{1, 2, \ldots, N\}$ represents a 3D scene with $N$ neural points. Each neural point has a learnable 3D position $\mathbf{p} \in \mathbb{R}^3$ and a learnable feature vector $\mu \in \mathbb{R}^d$. Given a ray $\mathbf{r}_j$ represented by a camera center $\mathbf{o}_j \in \mathbb{R}^3$ and a rays direction $\mathbf{d}_j \in \mathbb{R}^3$ as the query, PAPR learns $K \ll N$ attention weights for each of the $K$ nearest points around the ray. It then uses the attention weights to aggregate the value embedding vectors for the $K$ points to get a feature vector $\mathbf{f}_j$ that captures the color of the ray. To map the query and key into the same feature space, PAPR uses MLPs to learn embedding vectors:

$$\mathbf{q}_j = f_{\theta_Q}\left(\gamma\left(\mathbf{d}_j\right)\right), \quad \mathbf{k}_{ij} = f_{\theta_K}\left(\left[\gamma\left(\mathbf{h}_{i,j}\right), \gamma\left(\mathbf{t}_{i,j}\right), \gamma\left(\mathbf{p}_i\right)\right]\right), \tag{1}$$

where $f_\theta$ are the embedding MLPs, $\gamma$ is the positional encoding function applied to the inputs. $\mathbf{h}_{ij}$ and $\mathbf{t}_{ij}$ are two ray-dependent point feature vectors, computed as:

$$\mathbf{p}'_{ij} = \mathbf{o}_j + \langle \mathbf{p}_i - \mathbf{o}_j, \mathbf{d}_j \rangle \cdot \mathbf{d}_j, \quad \mathbf{h}_{ij} = \mathbf{p}'_{ij} - \mathbf{o}_j, \quad \mathbf{t}_{ij} = \mathbf{p}_i - \mathbf{p}'_{ij}, \tag{2}$$

where $\mathbf{p}'_{ij}$ is the position of the point's projection on ray $\mathbf{r}_j$. These features capture the point configuration of the neighborhood by incorporating the features from all $K$ points around a ray. To model ray-dependent appearance, PAPR learns a ray-dependent feature vector $\mathbf{v}_{ij}$ for each point:

$$\mathbf{v}_{ij} = f_{\theta_V}\left(\left[\gamma\left(\mathbf{h}_{i,j}\right), \gamma\left(\mathbf{t}_{i,j}\right), \gamma\left(\mu_i\right)\right]\right), \tag{3}$$

The ray's feature $\mathbf{f}_j$ can then be computed by $\mathbf{f}_j = \sum_{i=1}^{K} w_{ij}\mathbf{v}_{ij}$, where $w_{ij} = $ softmax$(\langle \mathbf{q}_j, \mathbf{k}_{ij} \rangle / \sqrt{d_\mathbf{k}})$ are the attention weights, $d_\mathbf{k}$ is the dimension of $\mathbf{k}_{ij}$. By spatially aggregating the feature vectors of all the rays from the same camera view, PAPR produces a feature

map of the view, which is then passed through a UNet-based renderer to predict RGB image $\hat{\mathbf{I}}$. The model is end-to-end learnable by minimizing a rendering loss between $\hat{\mathbf{I}}$ and ground truth $\mathbf{I}_{\mathrm{gt}}$:

$$\mathcal{L}_{\mathrm{render}} = \mathcal{L}_{\mathrm{MSE}}(\hat{\mathbf{I}}, \mathbf{I}_{\mathrm{gt}}) + \lambda \cdot \mathcal{L}_{\mathrm{LPIPS}}(\hat{\mathbf{I}}, \mathbf{I}_{\mathrm{gt}}) \tag{4}$$

which is a combination of mean squared error (MSE) and LPIPS metric (Zhang et al., 2018).

## 4 METHOD

While the main original motivation for PAPR was to learn a sparse point-based representation from scratch, we observe that a different property of PAPR is especially useful for enabling low-effort yet precise controls for editing.

### 4.1 3D GAUSSIAN SPLATTING VS. PAPR

Compared to 3DGS, a key difference is that in PAPR, points have no spatial extent. Rather than using splats to fill gaps between points, PAPR's attention mechanism outputs a set of weights over the $K$ nearest points for every ray. As attention weights are non-negative and sum up to 1, it essentially learns a convex interpolation of those points. Such interpolations are based on the local point configurations around the rays, which vary significantly across different rays observed during training. This variability enables PAPR to generalize effectively to unseen point sets after deformation.

With 3DGS, it is crucial to adjust the covariance of each affected Gaussian when editing, otherwise it may cause surface discontinuities. As shown in Fig. 2, fixing covariances can result in gaps between splats after editing. Moreover, even with adjusted covariances, it's challenging to get them right, as adjustments vary between splats. In contrast, PAPR lacks spatial extent around points, so no spatial adjustment is needed when editing—only the point locations must be changed.

Furthermore, even if the covariances were adjusted optimally, as shown in the left of Fig. 2, the shape surface after editing can still be not as smooth as it was originally. Neither shrinking nor rotating the splats would work since the former causes gaps, while the latter makes splats stick out. This is caused by the fact that non-rigid deformations of Gaussians are not necessarily Gaussian anymore. As a result, more Gaussian splats would need to be added at locations where the non-rigidity is the greatest. In contrast, since non-rigid deformations of a point set simply produce another point set, in PAPR, it suffices to move the points without adding new ones. The interpolator learned by PAPR should adapt to the changes in point positions, providing smooth and reliable interpolation for the updated point set, thereby preserving surface continuity.

### 4.2 SPARSE CONTROL POINTS

Even though PAPR was designed to learn a sparse point cloud, the number of points it requires is still too many for edits to be applied with low effort (i.e., 30,000 points). To manually modify the positions of every single point would be overly time-consuming and laborious, especially when the edit is complex and affects a large region of the scene. For example in Fig. 4, where the user attempts to stretch the chair back, editing PAPR either by parts or by points introduces artifacts or requires extensive manual adjustments. To address this issue, we develop a method to add sparse control points that can be used to guide the editing of PAPR's scene representation. As shown in Fig. 3, our method consists of three steps: (1) constructing a set of sparse control points, (2) reparameterizing each original point in terms of the control points, and (3) jointly finetuning the point coefficients, the control points and the neural renderer. As shown in Fig. 4, stretching the chair back involves dragging only a few control points with our method.

**Constructing Sparse Set of Control Points**  Given a pre-trained point cloud $\mathcal{P} = \{\mathbf{p}_i\}_{i=1}^N$ from PAPR, we construct a set of $M$ sparse control points $\mathcal{S} = \{\mathbf{s}_j\}_{j=1}^M$ to guide the deformation of the points in $\mathcal{P}$. Each sparse control point has a learnable 3D position $\mathbf{s}_j$, where $M \ll N$. The sparse control points $\mathcal{S}$ are only used to determine the positions of the points in $\mathcal{P}$ and won't be rendered. To get the sparse control points, one can either sample points from $\mathcal{P}$ using farthest point sampling (FPS) algorithm (Eldar et al., 1994), or train a point-based neural renderer with fewer points on the same scene and take the learned point cloud as the control points. We find that the difference between

Figure 4: Comparison of editing results between PAPR and our method. While editing PAPR by parts or points leads to visible artifacts or requires extensive adjustment, our method achieves the desired deformation—stretching the chair back—by adjusting only a few control points.

the two choices during the deformation and rendering process is negligible in our method. We use $M = 512$ in all our experiments following SC-GS.

**Reparameterizing Original Points via Control Points**   To enable control of the points in $\mathcal{P}$ using the control points in $\mathcal{S}$, we need to relate the position of each point $\mathcal{P}$ in the positions of the control points. We reparameterize the position $\mathbf{p}_i$ of each point as a linear combination of $k$ control points closest to it, we use $k = 5$ for all the experiments. To prevent $\mathbf{p}_i$ from moving erratically, we constrain the linear combination coefficients to be between 0 and 1, which helps the neural renderer to generalize to the deformed shapes. More concretely, given each point $\mathbf{p}_i$, we aim to find the coefficients $\beta_{i,j}$ that approximately satisfies:

$$\sum_{j \in \text{kNN}(\mathbf{p}_i, S)} \beta_{i,j} \mathbf{s}_j \approx \mathbf{p}_i, \text{ s.t. } \beta_{i,j} \in (0, 1) \tag{5}$$

To this end, we reparameterize $\beta_{i,j}$ as $\sigma(b_{i,j})$, where $\sigma(\cdot)$ denotes the sigmoid function and $b_{i,j}$ can take on any real value. We can then formulate an unconstrained optimization problem to find $\beta_{i,j}$'s that minimize the Euclidean distances between the parameterized and original points:

$$\min_{b_{1,1}, \dots, b_{N,M}} \sum_{i=1}^{N} \left\| \sum_{j \in \text{kNN}(\mathbf{p}_i, S)} \sigma(b_{i,j}) \mathbf{s}_j - \mathbf{p}_i \right\|_2^2 \tag{6}$$

**Joint Finetuning**   After finding the best coefficients $\beta_{i,j}$, we replace the original point positions $\mathbf{p}_i$ in the representation with the point positions reconstructed from $\beta_{i,j}$ and control points $\mathbf{s}_j$. We then finetune $\beta_{i,j}$, $\mathbf{s}_j$, and the neural renderer end-to-end using the rendering loss. As shown in Table. 2, the joint finetuning significantly improves the rendering quality of our model, ensuring better rendering quality after deformation. In the end, we obtain a neural rendering model whose scene representation is parameterized by the sparse control points.

### 4.3 Shape Editing through Control Points

With the finetuned model, where every point in the point cloud is parameterized by the control points, shape edits can be performed by simply adjusting the control points while keeping $\beta_{i,j}$ fixed. As a result, the points in the point cloud follow the movements of the control points, while all the other model parameters remain unchanged. A single forward pass through the model then transforms the deformed point cloud into the rendered image.

## 5 Experiments

### 5.1 Datasets and Baselines

To validate the effectiveness of our method in different editing scenarios, we compare PROSE with both geometry proxy-based and geometry proxy-free methods. For the proxy-based method we compare with Deforming-NeRF (Xu & Harada, 2022) and Mani-GS. Deforming-NeRF enables

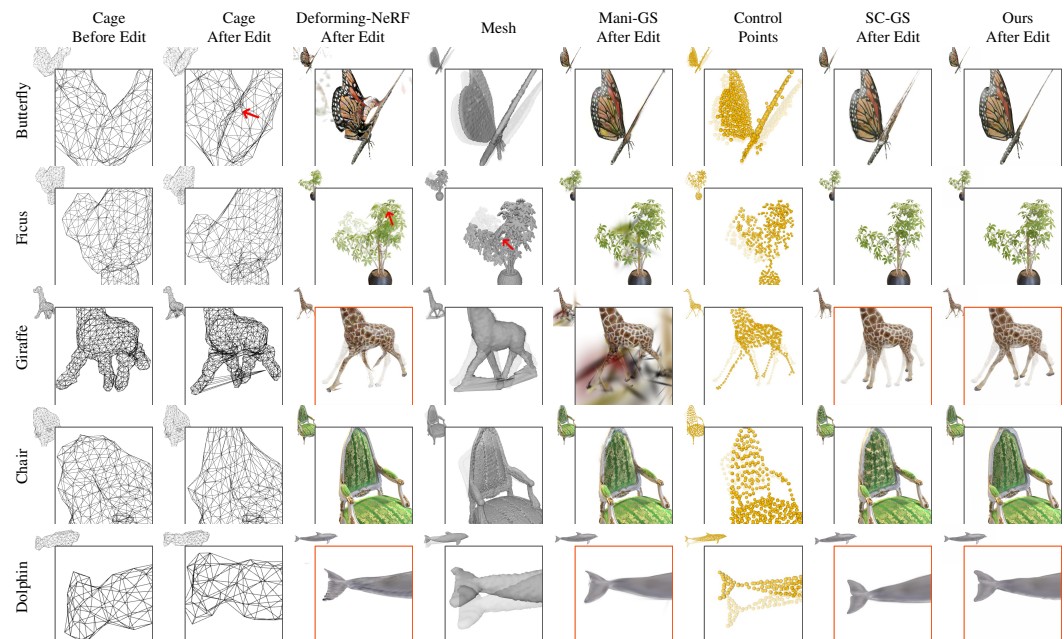

Figure 5: Qualitative comparison with proxy-based methods (Deforming-NeRF and Mani-GS) and a leading proxy-free method (SC-GS). We also visualize the geometry proxies or control points used by each method. As shown, the proxy-based methods face challenges in handling self-intersections (e.g., Butterfly) and part-level edits (e.g., Ficus and Giraffe) due to the connectivity constraints of the mesh representations. In contrast, proxy-free methods like SC-GS and our approach demonstrate greater robustness to a variety of deformations. Furthermore, our method consistently maintains surface continuity and avoids artifacts observed in SC-GS, such as holes in the back of the edited Chair and extruded Gaussians around the boundaries of Butterfly, Giraffe, Chair, and Dolphin.

deformations of the radiance field by editing a cage that encloses the object, and renders the edited scene by mapping the position and the view direction of the sampled points from the deformed space to the canonical space. Mani-GS (Gao et al., 2024b) is a 3DGS deformation method that edits a 3DGS representation by binding the Gaussians on a given mesh. For the proxy-free based method, we compare with SC-GS which supports editing 3DGS with sparse control points. Note that in all experiments for a fair comparison, we first train an SC-GS model, and then use the control points from SC-GS as the control points in our model. We fix the control points during joint-finetuning so that SC-GS and our method have the exact same control points when we deform them. We evaluate our method on the NeRF Synthetic dataset (Mildenhall et al., 2020), Tanks and Temples dataset (Knapitsch et al., 2017), and 5 dynamic objects from Objaverse (Deitke et al., 2022).

## 5.2 COMPARISON WITH PROXY-BASED BASELINES

Fig. 5 shows the qualitative comparison between PROSE and the baselines after performing various shape manipulations. For each method, we also visualize the geometry proxy or the control points used. To align the deformations across different methods for fairness, we first manually deform the cage from Deforming-NeRF, and then use the cage to deform the mesh or the control points of other methods through cage-based deformation. To ensure a fair comparison, we use the same set of control points to control the point clouds of our method and SC-GS.

As shown in the figure, the cage-based approach often suffers from self-intersections during editing, leading to incorrect deformation results, such as the artifacts in Butterfly. Additionally, a cage typically loses fine geometry details, causing edits to inadvertently affect unrelated parts of the model. For example, bending a branch of Ficus using the cage can result in the unintended deformation of a neighboring branch. Moreover, both cage and mesh-based methods struggle with part-level manipulations as they may fail to separate distinct parts. For example, as shown in the figure, if the two legs of the giraffe were initially close to each other before editing, they remain connected in both the cage and mesh representations. As a result, when the giraffe starts walking, the enforced

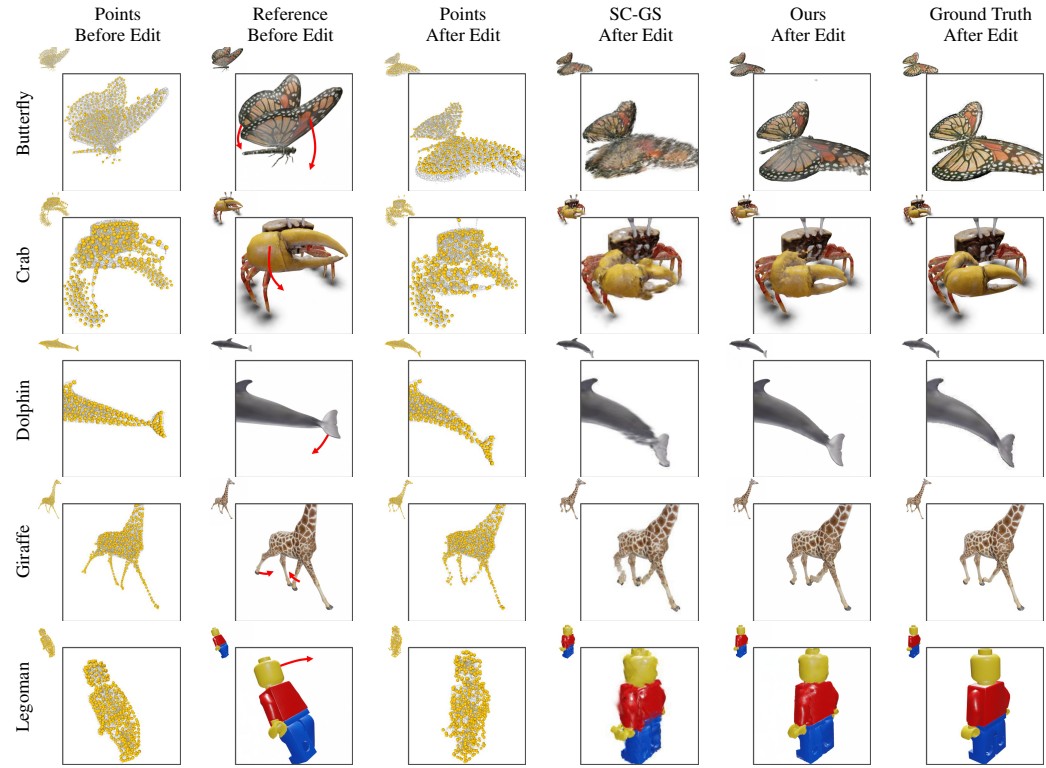

Figure 6: Qualitative comparison between SC-GS and our method on scenes from Objaverse.

connectivity leads to artifacts. Overcoming these issues requires extra effort, such as extracting a higher-quality mesh or manually editing the edges.

In contrast, SC-GS and our method do not rely on geometry proxies, making them more adaptable to various types of deformations. However, SC-GS often fails to preserve the surface continuity after editing, leading to issues such as holes in the back of the edited Chair and extruded Gaussians around the boundaries of Butterfly, Giraffe, Chair, and Dolphin. In contrast, our method consistently preserves surface continuity and avoids such artifacts across all the scenes.

## 5.3 COMPARISON WITH SC-GS

To further emphasize the advantages of using attention-based point representation over Gaussian splatting-based ones, we conduct additional comparisons with SC-GS.

Table 1: Quantitative evaluation of the rendering quality of our method and SC-GS after editing, on the dynamic scenes from Objaverse whose ground truth edited states are available.

| Metric | Method | Butterfly | Crab | Dolphin | Giraffe | Lego Man | Avg. | Improvement |
|---|---|---|---|---|---|---|---|---|
| PSNR ↑ | SC-GS | 16.77 | 20.33 | 29.43 | 26.27 | 21.22 | 22.80 | 25.4% |
| | PROSE (Ours) | **22.15** | **27.73** | **35.48** | **31.64** | **25.98** | **28.60** | |
| SSIM ↑ | SC-GS | 0.763 | 0.871 | 0.976 | 0.957 | 0.879 | 0.889 | 8.21% |
| | PROSE (Ours) | **0.927** | **0.962** | **0.994** | **0.987** | **0.939** | **0.962** | |
| LPIPS ↓ | SC-GS | 0.173 | 0.227 | 0.021 | 0.034 | 0.102 | 0.111 | 56.8% |
| | PROSE (Ours) | **0.063** | **0.098** | **0.006** | **0.015** | **0.060** | **0.048** | |

Table 2: Rendering quality before and after the joint finetuning before editing. As shown, the join finetuning stage improves the rendering quality of the model significantly.

| | NeRF Synthetic | | | Scenes from Objaverse (Unedited) | | | Tanks and Temples | | |
|---|---|---|---|---|---|---|---|---|---|
| | PSNR ↑ | SSIM ↑ | LPIPS ↓ | PSNR ↑ | SSIM ↑ | LPIPS ↓ | PSNR ↑ | SSIM ↑ | LPIPS ↓ |
| w/o joint finetuning | 28.80 | 0.945 | 0.047 | 22.89 | 0.920 | 0.060 | 25.77 | 0.882 | 0.119 |
| w/ joint finetuning | **32.30** | **0.959** | **0.036** | **34.76** | **0.988** | **0.010** | **29.39** | **0.920** | **0.089** |

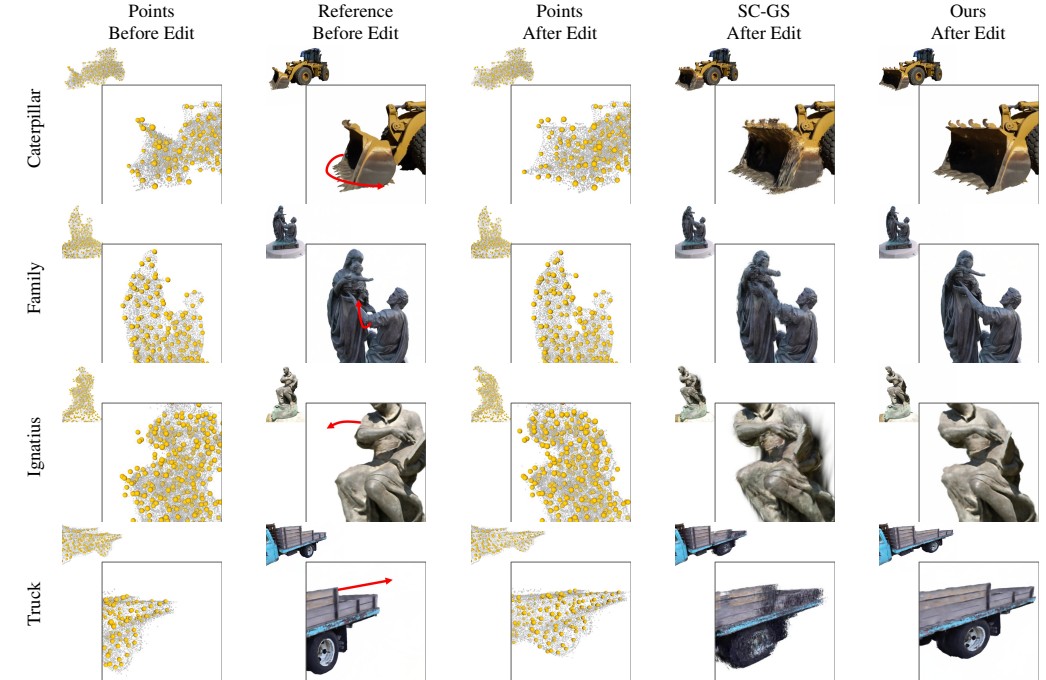

Figure 7: Qualitative comparison between SC-GS and our method on the Tanks and Temples dataset.

**Quantitative Comparison**  To quantitatively evaluate our method, we compare with SC-GS on 5 dynamic scenes from Objaverse (Deitke et al., 2022). Specifically, we extracted two frames from each dynamic scene, with one serving as the original unedited scene for training and the other serving as the edited ground truth for evaluation. We rendered 100 views of the unedited scene (for training) and 200 views of the edited scene (for evaluation). As shown in Fig. 6, non-rigid deformations are required to deform from the unedited to the edited scene. To evaluate the quality of our edits against the ground truth, we need to find the optimal adjustments for the control point positions so that the rendered images closely align with the ground truth edited scene. To this end, we optimize the control point positions for 6 epochs using the rendering loss on the images of the edited scene while keeping all the other parameters fixed. It's important to note that we never finetune anything other than the control points as we aim to evaluate the *zero-shot* editing quality, so it is prohibited to use the test images to optimize the renderer. For SC-GS, due to vanishing gradients, control point positions remain suboptimal even after optimization. Instead of using the suboptimal control points, we use the control points that are optimized using our method on the edited scene to guide the editing of SC-GS for fairness. Table 1 shows the quantitative results of zero-shot editing, where only control points are moved and nothing else is changed. As shown in the table, our method consistently outperforms SC-GS across different scenes across all metrics.

**Qualitative Comparison**  Fig. 6 shows the qualitative results using the optimized control points. As shown, SC-GS exhibits artifacts after editing, e.g., holes in Butterfly and Crab, and the misaligned Gaussian covariances in Dolphin, Giraffe, and Lego Man. In contrast, our method preserves both high-frequency details and surface continuity. To further demonstrate the effectiveness of our approach on real-world objects, we provide additional qualitative results on the Tanks and Temples dataset (Knapitsch et al., 2017) in Fig. 7. As shown, our method effectively preserves surface continuity and eliminates artifacts observed in SC-GS, such as holes and extruded Gaussians, even after non-rigid deformations.

## 6 CONCLUSION AND FUTURE WORK

In this paper, we present a method that enables shape editing of neural renderers with sparse controls, allowing for intuitive user editing while maintaining high-quality rendering. Unlike prior approaches, our method eliminates the need for fitting geometry proxies by leveraging attention-based neural point representations. By jointly optimizing rendering quality and point movements induced by sparse control manipulations, our approach prevents erratic deformations and achieves better rendering quality after editing. While we use a simple linear deformation as proof-of-concept, more sophisticated differentiable deformation techniques and multi-level control points could further enhance performance, which we leave for future work.

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
