# OpenReview forum: "PROSE: Point Rendering of Sparse-Controlled Edits to Static Scenes"
_ICLR.cc/2026/Conference — Submitted to ICLR 2026_

### Official Review · Reviewer_Y23b · 2025-10-30

**Soundness:** 3
**Presentation:** 4
**Contribution:** 3
**Rating:** 6
**Confidence:** 4

**Summary:**

This paper proposes PROSE, a method for performing shape editing of static 3D objects using sparse control points under a point-based neural renderer (PAPR) framework. The shape editing mechanism parameterizes all scene points as convex combinations of a small number of control points, and fine-tunes both control-point mappings and the renderer jointly. Experiments show that PROSE maintains rendering fidelity and surface continuity during deformation, outperforming both proxy-based and proxy-free baselines (e.g., SC-GS).

**Strengths:**

- This paper has clear motivation and problem definition, and the introduction is very informative to me, which concisely outlines limitations of mesh/cage-based editing and of Gaussian-based representations (e.g., discontinuities, self-intersections).
- The paper evaluates on NeRF Synthetic, Tanks and Temples, and Objaverse scenes. Qualitative results show superior deformations (continuous surfaces, fewer artifacts) compared with SC-GS and proxy-based methods (Deforming-NeRF, Mani-GS).

**Weaknesses:**

- The core components—PAPR and sparse control deformation—are both existing ideas, and the paper’s novelty lies primarily in combining them.
- While the method aims for “low-effort editing,” there is no user study or quantitative measure of editing convenience, leading to limited user-interaction perspective.
- The method's behavior under extreme deformations (e.g., stretching by >2× or bending >90°) is not shown, as well as the renderer’s generalization.

**Questions:**

- Why was a simple sigmoid-weighted linear combination chosen instead of e.g., RBF kernels or learned attention-based weighting similar to PAPR’s own mechanism?
- The re-parameterization of each point as a convex combination of k-nearest control points assumes local linearity. Have you observed any cases where this assumption fails, e.g., for topological changes or large-scale non-linear deformations?
- In PROSE, both the PAPR representation and the sparse control point mechanism seem essential to achieving smooth, proxy-free shape editing. Could you clarify whether these two components are fundamentally coupled? In other words, is the proposed sparse control framework specifically dependent on PAPR’s point-based attention interpolation, or could it, in principle, be applied to other point-based scene representations?

---

### Official Review · Reviewer_5tBK · 2025-10-31

**Soundness:** 1
**Presentation:** 3
**Contribution:** 1
**Rating:** 2
**Confidence:** 5

**Summary:**

This paper studies scene editing with human-indicated deformation. The proposed method is based on a point-based 3D representation, PAPR, that selects control points and reparameterizes the scene with a smaller number of points. PAPR's attention-based rendering natively supports various deformations with no gaps between 3DGS. Compared with the selected baselines, the proposed method achieves better performance.

**Strengths:**

- The method is straightforward and easy to understand.
- The results shown in this paper are good.
- Video is provided in supplementary materials to show that the proposed method can generate "dynamic" scenes.

**Weaknesses:**

- A crucial previous work is not discussed and compared in this work: NeuralEditor (CVPR'23).
    - NeuralEditor shares lots of similarities with this work: point-based scene representation, kNN-interpolation-based aggregation, point-guided deformation, etc. These should be discussed to emphasize the difference and superiority of the proposed method.
    - NeuralEditor also proposes a shape deformation benchmark based on the NeRFSynthetic dataset, with Blender files to help deform the control points, and ground truth edited scenes to evaluate PSNR/SSIM metrics. Given this is the only benchmark for scene deformation, it is necessary to report these numbers. Also, the deformation in this benchmark is more aggressive than the ones shown in the paper.
- One of the contributions of the paper is to reduce the points in PAPR to ~30K control points and apply interpolation-based reparameterization. However, this seems unnecessary, as those deformations can be applied to a huge (more than 1M) point cloud using Blender-style mesh deformation (manually indicate a space region with a closed mesh and deform the mesh to indicate the space deformation).
    - All of the edits in Fig.4 can be achieved by this Blender-style editing on a huge point cloud. "Edit by parts" can be achieved by selecting a part with a mesh and **move** it along the Z-axis direction; while "Edit by control points" only requires one to select a part with a mesh and then **stretch** it along the Z-axis direction, while keeping the bottom position of the mesh fixed.
    - All these can be simply operated by a human in Blender: import the point cloud of the scene, then add a simple mesh with a few vertices (e.g., 8 vertices for a cuboid) to indicate the deformed region, and finally move the vertices of the mesh to deform all the points inside.
    - Therefore, control points do not bring ease or enable new possibilities of editing.
    - On the other hand, ~30K points are still unable to be manually manipulated by humans "with low efforts", at least no lower than indicating the region in Blender with 8 vertices. Continuing to decrease the number of control points will also hurt the granularity of the possible editing.
- The proposed method is an incremental work based on the existing work, PAPR. Given that the design of control points is problematic (see previous point), the contribution of this paper seems minor.
- The attention-based rendering, though, helps fix the holes and gaps between the points after deformation, making the model unable to perform edits like "cutting shape in the middle" (e.g., Fig.4 "Edit by Parts") or the scene morphing supported by some of the previous work (NeuralEditor).
- The paper did not mention how to deal with the view-dependency issue, like mirror reflection. Therefore, the proposed method may fail in the deformation of scenes with these effects, e.g., NeRFSynthetic-Materials.

**Questions:**

- Could you please discuss the missing baseline (NeuralEditor), and report the numbers and results on its benchmark?
- To rebut the criticism on control points I mentioned in "weaknesses", could you please emphasize the advantages of them that cannot be achieved by the original PAPR representation? E.g., rendering/training efficiency?

---

### Official Review · Reviewer_31XM · 2025-11-02

**Soundness:** 3
**Presentation:** 3
**Contribution:** 2
**Rating:** 2
**Confidence:** 3

**Summary:**

This paper presents a point-based framework for sparse and interactive editing of static scenes. Instead of relying on Gaussian splats or geometry proxies, it adopts Proximity Attention Point Rendering (PAPR) to represent a scene as a point cloud. Each dense point is reparameterized by a small set of sparse control points (≈512), where every original point is expressed as a linear combination of its k-nearest control points using sigmoid-constrained weights. During editing, moving the sparse control points directly drives geometric deformation.

**Strengths:**

1. Proxy-free point-based design. This work represents scene with PAPR instead of Guassian splats or geometry proxies, resulting in smoother and more stable deformation.
2. Effective control mechanism. Construct a set of sparse control points to guid the scene editing, which improves the efficiency.
3. Experiments on NeRF Synthetic, Tanks & Temples, and Objaverse datasets demonstrate high-quality object editing.

**Weaknesses:**

1. Limited Novelty: This paper combines the scene representation method PAPR with sparse point control to improve the quality of edited results, which constitutes an incremental modification rather than a fundamental methodological innovation and raises the question of whether the significant improvement in metrics mainly stems from the use of a more suitable scene representation.
2. Scope misalignment: scene-level vs. object-level evaluation. Despite claiming to handle “static scene editing”, all experiments in this paper are performed on single object. There is no demonstration of full scene-level editing involving multiple objects, occlusions, or complex background.
3. Missing comparison with a previous method. Drag Your Gaussian (SIGGRAPH Asia 2025), which tackles a very similar goal—sparse control–driven editing, is omitted discussion.

**Questions:**

1. How's the performance compared with SOTA method (e.g., Drag Your Gaussian, SIGGRAPH Asia 2025)?
2. How scalable is PROSE to large, multi-object scenes with occlusions or complex background?
3. Efficiency: What are the GPU memory requirements and runtime for performing an edit in practice?

---

### Official Review · Reviewer_LokL · 2025-11-06

**Soundness:** 3
**Presentation:** 2
**Contribution:** 3
**Rating:** 6
**Confidence:** 4

**Summary:**

This paper aims to improve the quality and flexibility of 3D object editing by introducing a sparse control-point framework built upon a point-based 3D representation (PAPR). Each original 3D point is represented as a linear combination of its k-nearest control points, enabling local deformations to be expressed compactly and efficiently. The influence weights of the control points are optimized through backpropagation to achieve fine-grained and spatially coherent edits. Experimental results, both qualitative and quantitative, demonstrate that the proposed method outperforms SC-GS (a related approach that adopts a similar sparse control-point idea but operates on GS representation), in terms of editing quality and structural consistency.

**Strengths:**

+  Conceptually simple and effective: The method builds on a clear and intuitive idea—using sparse control points to guide deformation—which is easy to grasp yet powerful in practice.

+ Better experimental results: Both qualitative and quantitative evaluations demonstrate significant improvements over prior methods, (mainly compared with SC-GS),  validating the effectiveness of the proposed approach.

+ Clarity of presentation: The paper is easy to follow.

**Weaknesses:**

- The main weakness of this paper lies in its limited technical novelty and insufficient analysis of its core contribution. Its sparse point control and kNN association between each original point and its control points, have already exist in SC-GS. The claimed difference (technical novelty aspect) mainly lies in the optimization of control-point weights, yet this aspect is not thoroughly validated. The paper lacks rigorous ablation studies to separate the effects of individual design choices, such as keeping the GS representation while applying the proposed sparse point control editing, or replacing SC-GS's representation with PAPR  for a more rigid, fair comparison. As a result, it remains unclear whether the reported improvements truly arise from the new formulation or simply from adopting the point-based (PAPR) representation. The current version feels more like an A + B combination of both ideas (SC-GS + PAPR).

- The overall paper structure also requires substantial improvement. Too much space is devoted to explaining why the point representation by PAPR is better than GS, which belongs to prior work rather than the authors’ own contribution. These details could be summarized conceptually in the main paper (e.g., supported by a single figure) and moved to the supplementary material. The main text should instead focus more on clarifying the differences in the sparse control-point design compared with SC-GS, and include more meaningful ablation studies to support these claims. Strengthening this focus would make the contribution clearer and the paper more convincing.

- The paper also lacks in-depth quantitative experiments to substantiate its claims. Beyond the previously mentioned need for clearer ablation studies in weakness 1, several analyses are missing. For example, there is no evaluation of joint finetuning versus optimizing only the control-point coefficients, leaving it unclear how much each contributes to performance. The paper should further investigate the effects of the number of sparse control points and the choice of k in the kNN association, as these factors directly influence editing quality and computational cost. Additionally, it would be valuable to evaluate nvs consistency under editing, to show whether the edited geometry remains coherent and photometrically consistent across different viewpoints. Also, testing on more challenging dynamic scenes, such as the Tyrannosaurus Dance sequence widely used in NeRF-based benchmarks, would strengthen the evidence for robustness and generalization.

-There is no clear study on editable boundaries or failure cases, which is essential to understand the method’s limitations and applicability.

**Questions:**

- Figure 8 mismatch: seems the caption and textual description of Figure 8 appear inconsistent with the visual content？

- Evaluation scope (Table 3): In Table 3, how many objects were selected from the Objaverse dataset for evaluation? Please specify the selection criteria and whether the chosen subset represents the dataset’s diversity.

- Interactive editing interface: Will the authors provide or release an interactive editing interface?
- Computation and time cost: What's the time cost of each step in their pipeline？

---

### Meta-Review · Area_Chair_oLg4 · 2026-01-06

**Summary:**

The reviewers acknowledged the paper's goal of enabling proxy-free shape editing using sparse control points and a point-based neural renderer (PAPR). However, significant concerns regarding technical novelty and the practical utility of the proposed workflow were raised. The methodology is primarily viewed as an incremental combination of existing techniques (SC-GS and PAPR). Furthermore, the paper omitted comparisons with critical baselines, and the authors did not submit a rebuttal to address these challenges. Given the lack of author engagement and the unaddressed technical gaps, I recommend rejection.

**Reviewer Concerns:**

NA. The authors did not submit the rebuttal.

**Reviewer Scores:**

NA

---

### Decision · Program_Chairs · 2026-01-26

Reject